# Faricimab Reverts VEGF-A_165_-Induced Impairment of the Barrier Formed by Retinal Endothelial Cells

**DOI:** 10.3390/ijms26094318

**Published:** 2025-05-01

**Authors:** Dominik M. Jung, Isabell Fuezy, Lyubomyr Lytvynchuk, Heidrun L. Deissler

**Affiliations:** 1Department of Ophthalmology, Justus Liebig University Giessen, 35392 Giessen, Germany; 2Department of Ophthalmology, University Hospital Giessen and Marburg GmbH, Campus Giessen, 35392 Giessen, Germany; 3Karl Landsteiner Institute for Retinal Research and Imaging, 1030 Vienna, Austria

**Keywords:** retinal endothelial cells, VEGF-A, angiopoietin-2, anti-VEGF drugs, faricimab, cell index measurement, tight junction, claudin-1, claudin-5, plasmalemma vesicle-associated protein

## Abstract

VEGF-A_165_-induced persistent dysfunction of the barrier formed by immortalized bovine retinal endothelial cells (iBREC) is only transiently reverted by inhibition of VEGF-A-driven signaling. As angiopoietin-2 (Ang-2) enhances the detrimental action of VEGF-A_165_, we studied if binding of both growth factors by the bi-specific antibody faricimab sustainably reverts barrier impairment. Confluent monolayers of iBREC were exposed to VEGF-A_165_ for one day before 10–1000 µg/mL faricimab was added for additional five days. To assess barrier function, we performed continuous electric cell–substrate impedance, i.e., cell index, measurements. VEGF-A_165_ significantly lowered the cell index values which recovered to normal values within hours after the addition of faricimab. Stabilization lasted for two to five days, depending on the antagonist’s concentration. As determined by Western blotting, only ≥100 µg/mL faricimab efficiently normalized altered expression of claudin-1 and claudin-5, but all concentrations prevented further increase in plasmalemma vesicle-associated protein induced by VEGF-A_165_; these proteins are involved in barrier stability. Secretion of Ang-2 by iBREC was significantly higher after exposure to VEGF-A_165_, and strongly reduced by faricimab even below basal levels; aflibercept was significantly less efficient. Taken together, faricimab sustainably reverts VEGF-A_165_-induced barrier impairment and protects against detrimental actions of Ang-2 by lowering its secretion.

## 1. Introduction

Vascular endothelial growth factor-A (VEGF-A)_165_ elevates the permeability of the endothelial cell (EC) layer of retinal microvessels, eventually resulting in vision-threatening macular edema, a hallmark of burdening retinal diseases such as age-related macular degeneration, diabetic macular edema, retinal vein occlusion, and myopic retinopathy [1,2,3]. In vitro, VEGF-A_165_ induces barrier impairment of primary and immortalized retinal endothelial cells (REC) isolated from various species, including human, through activating VEGF receptor 2 (VEGFR2)-driven signaling [2,4,5,6,7].

The function of the barrier formed by microvascular EC, e.g., REC, can reliably be assessed by continuous electric cell–substrate impedance measurements with a microelectronic biosensor system for cell-based assays, i.e., determination of the so-called cell index [8,9,10]. A high cell index reflects a tight barrier and vice versa. A few hours after the addition of VEGF-A_165_ to a confluent monolayer of immortalized bovine REC (iBREC), the cell index values start to decline and remain low for at least several days [10,11,12,13]. A similar severe impairment of the barrier has also been observed for human REC (huREC) [14]. Interfering with VEGF-A-mediated signaling through binding of the growth factor by VEGF antagonists ranibizumab and brolucizumab efficiently, but only transiently, reverts VEGF-A_165_-induced changes: low cell index values are normalized within one day but start to decline during extended incubation for up to five days [12,15,16]. Inhibition of VEGFR2 completely normalizes low cell index values only when inhibitors were added one day after exposure to the growth factor but not on day three [5,13]. Therefore, the modulation of VEGF-A_165_’s action by other growth factors or cytokines is likely, and the angiogenic growth factor angiopoietin-2 (Ang-2) seems to be a promising candidate. Ang-2 is secreted by iBREC during prolonged exposure to VEGF-A_165_ although at much lower concentrations compared to other microvascular EC [17,18,19,20]. Studies using a rabbit retina hyperpermeability model showed that concentrations of Ang-2 in the vitreous are also strongly elevated after intravitreal injection of VEGF-A_165_, significantly prevented by the VEGF-binding proteins aflibercept, brolucizumab and ranibizumab [21,22]. Ang-2 stimulates the proliferation of iBREC, but does not impair their very tight barrier [17]. The growth factor rather significantly enhances the detrimental action of VEGF-A_165_, and this disturbance is efficiently prevented by blocking VEGF-A-mediated signal transduction [17]. On the cellular level, the extended exposure of iBREC to VEGF-A_165_ over several days changes the expression of proteins regulating para- and transcellular flow, also observed for primary bovine and human REC. Amounts of tight junction (TJ)-protein claudin-1 are persistently reduced, those of TJ-protein claudin-5 remain stable early after the addition of the growth factor but increase later on [2,5,6,7,10,12,13,17]. The key regulator of transcellular flow, plasmalemma vesicle-associated protein (PLVAP), is hardly expressed by unchallenged REC, but its expression is dramatically enhanced after exposure to VEGF-A_165_ [12,23,24,25]. VEGF-A antagonists brolucizumab and ranibizumab completely reverse VEGF-A_165_-induced changes in claudin-1 and claudin-5, but only partly those in PLVAP [12].

Using our well-established model of immortalized microvascular EC of the bovine retina (iBREC), we now investigated if dual binding of both growth factors VEGF-A and Ang-2 by the bi-specific IgG faricimab efficiently and sustainably reverses VEGF-A_165_-induced changes in the barrier formed by these cells, i.e., low cell index values as a measure of barrier function, and altered expression of claudin-1, claudin-5, and PLVAP [11,20]. In order to mimic patients’ situation, we focused on the extended treatment of the cells with the effectors for up to six days. Although of non-human origin, primary and immortalized microvascular EC of the bovine retina represent an authentic in vitro model of the highly impermeable barrier formed by REC in vivo [7,11,24]. Their monolayers establish a very strong barrier reflected by persistently high values of the transendothelial electrical resistance or the cell index, accompanied by strong expression of TJ-protein claudin-5 typically expressed by microvascular EC [2,7,10,11,12,13,26]. In contrast to macrovascular EC, levels of PLVAP are extremely low [17,24]. The homogenous iBREC cell line can be stimulated by human growth factors, and, most importantly, is free of contaminating cells of other types often found in primary cultures which might affect the accuracy of in vitro studies [2,11,17].

## 2. Results

### 2.1. General Information

All experiments to investigate the changes induced by VEGF-A_165_ and subsequent treatment with faricimab were performed at least thrice, always with confluent monolayers of iBREC. To ensure maintenance of their typical phenotype, cells were cultured in cell culture medium adapted to their special needs [27]. After establishing a tight monolayer, recombinant human VEGF-A_165_ (a final concentration of 50 ng/mL) was added to the cells to induce a dysfunction of the barrier. One day later, faricimab (final concentrations of 10 µg/mL, 100 µg/mL, or 1 mg/mL) was placed in the cell culture medium; thus, VEGF-A_165_ was present until the end of the experiment [12,13]. In control experiments, cells were processed in exactly the same way without studied effector(s). Investigated concentrations of faricimab can easily be achieved by intravitreal injections [28].

### 2.2. Faricimab Strongly Suppressed Higher Secretion of Ang-2 by VEGF-A_165_-Treated iBREC

The extended cultivation of cells over several days could lead to cellular stress; therefore, we determined if long-term-cultured iBREC secreted interleukin (IL)-6, a marker of cellular stress [27,29]. However, the cytokine was not detected by ELISA in supernatants obtained from confluent iBREC monolayers cultured for up to nine days, unaltered by exposure to 50 ng/mL VEGF-A_165_; values were below the minimal detectable dose of 78 pg/mL. Therefore, the chosen conditions of cultivation support the establishment and maintenance of a monolayer of healthy microvascular endothelial cells.

Faricimab binds strongly to its target protein, VEGF-A_165_, and to confirm that this complex is still stable after the extended treatment of VEGF-A_165_-challenged iBREC with faricimab for five days, we determined the amounts of non-complexed VEGF-A in supernatants and cell extracts. Faricimab and the detection antibody of the used ELISA likely bind to the same region of the growth factor; therefore, only unbound VEGF-A can be measured [20]. A four-fold molar excess of faricimab over VEGF-A_165_ indeed prevents the detection of the growth factor by the used ELISA [30]. As anticipated, relevant amounts of the growth factor were measured in cell culture supernatants and cell extracts of iBREC exposed to the growth factor for six days (supernatants: 19 ± 6 ng/mL VEGF-A; N = 24; cell extracts: 0.25 ± 0.11 ng/10^6^ cells VEGF-A; N = 6). However, signals obtained from supernatants or extracts of cells additionally exposed to faricimab (final concentrations: 10 µg/mL, 100 µg/mL, and 1 mg/mL) were below the minimal detectable dose of 9 pg/mL, and this did not depend on the antagonist’s concentration.

In addition, Western blot analyses of cell extracts confirmed the presence of internalized faricimab after exposure to the growth factor and subsequently to the antagonist for as long as five days (Figure 1). However, significantly less faricimab was detected when cells had been treated with the lowest concentration of 10 µg/mL faricimab.

We also studied whether faricimab changed secretion of Ang-2 by VEGF-A-challenged iBREC. Faricimab does not interfere with binding of the detection antibody to Ang-2 of the used ELISA, i.e., free Ang-2 and Ang-2/faricimab complexes are detected [17]. Low concentrations of ~250 pg/mL Ang-2 were measured in the supernatant of unchallenged iBREC, and levels were significantly higher three and six days after the addition of VEGF-A_165_ (Figure 2) [17]. The increased secretion of Ang-2 was efficiently lowered by subsequent exposure to the VEGF antagonist aflibercept (Figure 2). Most interestingly, faricimab reduced Ang-2 levels much more noticeably, even significantly below basal levels (Figure 2). Relevant differences between both time points were not observed.

### 2.3. Faricimab Efficiently Reverted VEGF-A_165_-Induced Barrier Impairment

To assess the capability of faricimab to efficiently and sustainably revert the VEGF-A_165_-induced barrier impairment, we measured the cell index of an iBREC monolayer. After establishing a tight barrier as indicated by high cell index values of ~20, the growth factor was added. One day later, the antagonist was placed in the cell culture medium, and the cell index was continuously monitored for up to six days (Figure 3).

The growth factor induced a persistent reduction in the cell index values indicative of impairment of the cells’ barrier (Figure 3a). After the addition of faricimab at a final concentration of 10 µg/mL to VEGF-A_165_-pretreated iBREC, the cell index values increased and reached normal values twelve hours later (Figure 3b). Cell index values remained similar to those obtained for control cells for about two days, before they started to decline again. They were then significantly lower compared to those of the control cells but still significantly higher compared to those of VEGF-A_165_-exposed iBREC (Figure 3b). At a concentration of 100 µg/mL, faricimab reverted VEGF-A_165_-lowered cell index values also within hours; cell index values remained close to those of the control cells even late after the antagonist’s addition (Figure 3c). Increasing the concentration of faricimab to 1 mg/mL also resulted in rapid reversion of VEGF-A_165_-reduced cell index values, and normalized cell index values did not differ from those of control cells even during extended incubation for five days (Figure 3d). Accordingly, faricimab at a concentration of 1 mg/mL was significantly more efficient compared to 10 µg/mL (Figure 3e) or 100 µg/mL (Figure 3f).

### 2.4. Faricimab Reverted VEGF-A_165_-Induced Changes in the Expression of Claudin-1 and PLVAP

Constantly low amounts of the TJ-protein claudin-1 correlate with VEGF-A_165_-induced barrier impairment whereas the expression of TJ-protein claudin-5 is significantly higher only after extended exposure to the growth factor, and that of adherens junction (AJ) protein vascular endothelial cadherin (VEcadherin) remains largely unchanged [2,10,12,13,31]. To investigate if faricimab sustainably normalized VEGF-A_165_-induced changes, we assessed the expression of these candidate proteins (Figure 4).

As anticipated, low amounts of claudin-1 (Figure 4a,e) were observed after exposure to VEGF-A_165_ for one, three, or six days. When cells had been treated with VEGF-A_165_ for one day and subsequently with faricimab for only two days, the expression of claudin-1 was efficiently re-instated independent on the antagonist’s concentration (Figure 4a). However, after the extended treatment of VEGF-A_165_-exposed iBREC with 10 µg/mL faricimab, the expression of claudin-1 was significantly lower at this later time point (relative expression of claudin-1: 0.77 ± 0.27 on day 6 (N = 9) compared to 1.47 ± 0.64 on day 3 (N = 9); *p* = 0.014) (Figure 4a, left panel). Although the expression of claudin-1 was still significantly higher than that observed after exposure to VEGF-A_165_ only, it was now significantly lower compared to that of control cells (Figure 4a, left panel). In contrast, during prolonged exposure of VEGF-A_165_-challenged iBREC with ≥100 µg/mL faricimab, the expression of claudin-1 remained stable, and was significantly higher compared to that of VEGF-A-exposed cells (Figure 4a, middle and right panels).

Only after treatment with VEGF-A_165_ for six days, significantly more TJ-protein claudin-5 was expressed, but not when cells were also exposed to faricimab (Figure 4b,e). Extended exposure of VEGF-A_165_-pretreated cells to 10 µg/mL faricimab reduced expression of claudin-5 (relative expression of claudin-5: 0.82 ± 0.21 on day 6 (N = 10) compared to 1.41 ± 0.46 on day 3 (N = 9); *p* = 0.004); it was also significantly lower compared to that of control cells (Figure 4b, left panel). The expression of VEcadherin remained mostly stable; it was only significantly lower when VEGF-A_165_-exposed cells had also been treated with 1 mg/mL faricimab for additional five days (Figure 4c, right panel).

The expression of the regulator of intracellular transport PLVAP by iBREC is induced by VEGF-A_165_. One day after the addition of the growth factor, it was slightly but significantly higher (values of relative expression of PLVAP were: 0.34 ± 0.22 (N = 4) for control compared to normalized signals of VEGF-A_165_ of 1.00 (N = 4), *p* = 0.009). Even stronger signals were observed after the treatment of the cells for three or six days (Figure 4d). Interestingly, VEGF-A-induced higher PLVAP expression was efficiently counteracted by faricimab only after prolonged treatment regardless of its concentration (Figure 4d,f).

Taken together, faricimab efficiently reverted VEGF-A_165_-changed expression of proteins involved in the regulation of barrier stability, although significant deviations from normal expression levels were observed, especially late after the addition of the growth factor and its antagonist.

## 3. Discussion

VEGF-A_165_ persistently impairs the barrier formed by a monolayer of primary and immortalized microvascular retinal endothelial cells [2,5,6,7,10,12,13]. Since the blocking of VEGF-A-driven signal transduction by complexing the growth factor with VEGF-binding proteins or by inhibiting the tyrosine kinase activities of the growth factor’s receptor only transiently revert VEGF-A_165_-induced barrier impairment, we studied if additional binding of Ang-2 by faricimab is superior, using our well-established in vitro model iBREC [10,12,13,20]. In contrast to commercially available HuREC, iBREC and primary bovine REC (BREC) express very low if any PLVAP which regulates transcellular flow and is up-regulated by microvascular endothelial cells only under pathological conditions [12,13,23,24]. Accordingly, monolayers of primary and immortalized BREC give rise to high values of the transendothelial electrical resistance or the cell index persisting even over several days, and these values are typically much higher than those obtained with commercially available HuREC [17,18,25,32]. In contrast to co-culture models, interaction of REC with retinal pericytes cannot be taken into account by cultivating only the iBREC, but this disadvantage can be compensated at least in part by the usage of optimized cell culture media [7,19,33]. Taken together, iBREC represent a very reliable and more authentic model of the tight retinal–blood barrier, because similar to BREC, this cell line establishes a very tight barrier [5,12,13,17,24]. In spite of the non-human nature of the used cell model, it is a reasonable assumption that humanized IgG faricimab, which competes with binding of *human* Ang-2 to its receptor Tie2, also efficiently complexes *bovine* Ang-2 [20]. The overall identity and similarity of the human and bovine homologues of Ang-2 are extremely high, and—most importantly—the amino acid sequences of the receptor-binding regions of human and bovine Ang-2 are identical [34].

Aflibercept normalized extracellular Ang-2 elevated by VEGF-A_165_ treatment of iBREC, and our finding is in accordance with previously published observations using a rabbit retina hyperpermeability model [22]. Dual binding of both growth factors by faricimab was even more potent, suggesting that besides VEGF-driven signaling, the Tie2 pathway is also involved in down-regulating the secretion of Ang-2 [35]. The growth factor likely acts through an internal autocrine mechanism, in iBREC obviously efficiently counteracted by the internalized antagonist [36]. In this context, it is of interest that intravitreal injection of faricimab, but not of aflibercept, lowers the concentration of unbound Ang-2 in aqueous humor of patients with diabetic macular edema [37].

The modulation of VEGF-A_165_ detrimental changes in the iBREC barrier are observed only with Ang-2 concentrations which are an order of magnitude higher compared to those secreted by iBREC; Ang-2 on its own does not affect the stability of the cells’ barrier [17]. Nevertheless, one might speculate that dual binding of both growth factors is superior to reinstate and, even more importantly, maintain a functional barrier that was compromised by pre-treatment with VEGF-A_165_. Indeed, high concentrations of faricimab (≥100 µg/mL) efficiently reverted changes induced by the growth factor, including normalization of cell index values and the expression of proteins involved in barrier stabilization, e.g., claudin-1 and claudin-5 [2,5,6]. It is most likely that then both TJ-proteins are part of stable TJ complexes at the plasma membrane. Interestingly, iBREC exposed to faricimab only also express slightly more (≤1.5×) claudin-1 and claudin-5 after prolonged treatment with the antagonist, an observation also made for the VEGF-binding protein ranibizumab [17,38]. However, the up-regulation of the TJ-proteins by iBREC exposed to faricimab or ranibizumab for several days does not translate into an altered function of the cells’ barrier indicated by unchanged cell index values [17,38]. In accordance with our previous findings, the expression of the AJ-protein VEcadherin remained largely unchanged by the effectors, although its amounts were significantly lower after the exposure of the cells to VEGF-A_165_ followed by a high dose of the pharmacological formulation of faricimab, i.e., Vabysmo [12,13]. It is a reasonable assumption that the surfactant polysorbate-20—a component of the formulation—is responsible for the observed altered expression of VEcadherin, because it is significantly reduced by ≥0.0002% polysorbate-20 [38]. Surfactants prevent destabilization and precipitation of proteins, but they can be subject to modification and degradation by cellular enzymes [39]. However, low amounts of VEcadherin obviously do not necessarily correlate with a dysfunctional barrier, because the cell index values are still high [38]. Similar to our observations with VEGF-A-binding proteins ranibizumab or brolucizumab, the lowest faricimab concentration of 10 µg/mL only transiently reinstated low cell index values [12,13]. Subtle but significant deviations from the normal expression patterns of TJ-proteins claudin-1 and claudin-5 might account for the observed barrier impairment, because their reduced expression correlates with a dysfunctional barrier [2,10,27]. Whether or not these subtle changes result from residual VEGF-A-driven signaling is an interesting speculation. Relevant amounts of unbound intra- or extracellular VEGF-A were not detected even late after the antagonist’s addition, suggesting that the growth factor is very likely completely bound by faricimab present in an excess. However, complexing the growth factor by cellular proteins, e.g., (soluble) VEGF receptors, cannot be ruled out, and according to the manufacturer’s instructions, these complexes would not be measured by the used ELISA. Even low amounts of VEGF-A bound to the VEGF receptors—either at the cell surface or intracellular—could initiate or maintain VEGF signaling, eventually leading to barrier impairment [4,40].

The low expression of the regulator of transcellular flow PLVAP is a key feature of microvascular cells establishing a tight barrier, and its increased expression is associated with higher permeability of the endothelial layer [24,25]. De novo synthesis of PLVAP is induced by VEGF-A_165_, and PLVAP mRNA and protein can be detected at the earliest after one to two days [12,23,24,41,42]. Faricimab prevented further up-regulation of PLVAP by VEGF-A_165_-treated iBREC, but—likely due to a slow turn-over of the protein—this was achieved only late after the antagonist’s addition. PLVAP was still strongly expressed by VEGF-A_165_-challenged cells exposed to faricimab for only two days despite a closed barrier, as indicated by high cell index values. Therefore, the proper function of the iBREC barrier appears to be predominantly determined by the balanced expression of TJ-proteins, i.e., claudin-1 and claudin-5, and consequently, by restricted paracellular flow. Dual inhibition of VEGF-A and Ang-2 counteracts VEGF-A_165_-induced PLVAP expression to a greater extent than blocking VEGF-A alone with ranibizumab, as previously shown, suggesting a direct or indirect role of Ang-2 in the regulation of PLVAP expression [12,42].

Taken together, faricimab efficiently reverts the VEGF-A_165_-induced dysfunction of the barrier formed by an iBREC monolayer, and this stabilization can be maintained for several days with high concentrations of the antagonist. Of course, conclusions drawn from in vitro experimentation based on models with certain limitations cannot be transferred directly to the clinical assessment of patients and their used therapy, but, nevertheless, our findings may at least partially translate to the in vivo situation of (diabetic) macular edema caused by increased permeability of the retinal capillary endothelial cell monolayer due to VEGF-A_165_ present in the vitreous humor [3,43]. The pathogenesis of diabetic macular edema is also accompanied by elevated concentrations of Ang-2 in the vitreous humor, and as mentioned above, the growth factor strengthens the detrimental effects of VEGF-A_165_ in vitro [17,20,44,45,46]. Accordingly, clinical studies showed substantially maintained improvement of visual acuity and central subfield thickness by intravitreal injections of faricimab [28]. That faricimab strongly down-regulates secretion of Ang-2 in vitro as well as in vivo, suggests an additional protective role of the antagonist in early and late stages of diabetic retinopathy, e.g., by counteracting Ang-2-caused loss of retinal pericytes or Ang-2-stimulated retinal neovascularization [37,47].

## 4. Materials and Methods

### 4.1. Antibodies and Reagents

Information on antibodies used in this study is summarized in Table 1. The bi-specific antibody faricimab (Vabysmo; 120 mg/mL in 3.1 mg/mL L-histidine, 1.044 mg/mL L-methionine, 1.46 mg/mL NaCl, 54.8 mg/mL D-sucrose, 0.04% polysorbate-20, pH 5.5) and the Fc fusion protein aflibercept (Eylea; 40 mg/mL aflibercept in 10 mM sodiumphosphate, 40 mM NaCl, 0.03% polysorbate-20, 5% sucrose, pH 6.2) were purchased from Roche Pharma AG (Grenzach-Wyhlen, Germany) and Bayer AG (Berlin, Germany), respectively [20,21]. Recombinant human *Sf21*-expressed VEGF-A_165_ (#293VE) was bought from bio-techne (Wiesbaden, Germany), dissolved and stored as described [17].

### 4.2. Cultivation of iBREC

Telomerase-immortalized microvascular endothelial cells from bovine retina (iBREC)—established in our laboratory—were cultivated on surfaces coated with fibronectin (Corning, Amsterdam, The Netherlands) in Endothelial Cell Growth Medium MV (ECGM; #C-22120, Promocell, Heidelberg, Germany) containing 1 g/L glucose, 0.4% Endothelial Cell Growth Supplement/H, 90 µg/mL heparin, 10 ng/mL human epidermal growth factor (hEGF), 100 nM hydrocortisone, 5% fetal bovine serum (FBS; all supplements were from Promocell), and 300 µg/mL geneticin (Thermo Fisher Scientific, Schwerte, Germany) as described in great detail elsewhere [2,5,10,11,12,13]. Characterization of iBREC which were used from passages 25 to 50 counting from the stage of primary culture, and their responses to stimulation with human growth factors have been described in great detail [2,10,11,12,13].

### 4.3. Cell Index Measurements

The cell index (CI) was determined to assess the stability of the barrier formed by a monolayer of iBREC by electric cell–substrate impedance measurements using the microelectronic biosensor system for cell-based assays xCELLigence RTCA DP (Agilent, OLS, Bremen, Germany) as previously described [5,10]. Briefly, cells were cultured in ECGM-1 (same as ECGM lacking hEGF, but containing 1 µg/mL fibronectin) until a confluent cell monolayer was reached, indicated by a constantly high cell index (CI ~ 20) about four days later. Then, the cell culture medium was replaced by ECGM-1, and after one day, VEGF-A_165_ (final concentration of 50 ng/mL) was added and cells were incubated for another 24 h. Faricimab (final concentration of 10 µg/mL, 100 µg/mL, or 1 mg/mL) was placed in the cell culture medium, and the cell index was measured regularly every five minutes until the end of the experiments three or six days later. Recorded cell index values were normalized in relation to those measured immediately before the addition of VEGF-A_165_ (RTCA Software Pro, Version 2.6.1, Agilent), and the results were converted to graphs showing means and standard deviations with GraphPad Prism 9.4.1 (GraphPad Software, Boston, MA, USA) [10,12,17]. At the end of each experiment, the integrity of the confluent monolayer was confirmed by microscopy, cell culture supernatants were collected, and cells were harvested for the preparation of cell extracts [17]. iBREC were also cultured in fibronectin-coated T25-cell culture flasks (Sarstedt, Nuembrecht, Germany), treated in a similar manner as described above for harvesting cell culture supernatants and cells [17]. Samples were stored at −80 °C for further analyses.

### 4.4. Measurement of VGEF-A_165_, Ang-2 and IL-6 by ELISA

Confluent iBREC were pre-treated with 50 ng/mL VEGF-A_165_ for one day before either faricimab or aflibercept (final concentrations: 10 µg/mL or 100 µg/mL) were added, and cell culture supernatants were harvested two and five days after the addition of the inhibitors. Possibly secreted Ang-2 was determined in undiluted cell culture supernatants using Angiopoietin-2 Quantikine ELISA Kit (#DNAG20, bio-techne). To determine non-complexed VEGF-A in cell culture supernatants or cell extracts of iBRECs exposed to 50 ng/mL VEGF-A_165_ and subsequently to faricimab (final concentrations: 10 µg/mL, 100 µg/mL, and 1 mg/mL) for five days, we used the Quantikine ELISA VEGF-A Immunoassay Kit (#DVE00, bio-techne). The supernatants of cells treated only with VEGF-A_165_ and cell extracts were diluted 1:100 in phosphate-buffered saline without Ca^2+^ and Mg^2+^ (Thermo Fisher Scientific); all other supernatants were not diluted [12]. The marker of cellular stress IL-6 was measured with the IL-6 bovine uncoated ELISA Kit (#ESS0029, Thermo Fisher Scientific) in undiluted cell culture supernatants of iBREC cultured in ECGM-1 with or without 50 ng/mL VEGF-A_165_ for up to nine days [27,29]. Samples were processed at least in duplicates according to the manufacturers’ instructions, and analyte-dependent absorbance was measured at 450 nm (reference wavelength: 570 nm) 15–20 min after the addition of the stop solution, as described previously [12,17,27]. Standard curves for Ang-2 (0 to 3000 pg/mL; minimal detectable dose: 8 pg/mL), VEGF-A (0 to 1250 pg/mL; minimal detectable dose: 9 pg/mL), or IL-6 (0 to 5000 pg/mL; minimal detectable dose: ≤78 pg/mL) were always generated in parallel to the analyses of samples. Ang-2-specific signals were normalized to those obtained from control cells, and results are presented as scatter plots also showing means ± standard deviations.

### 4.5. Western Blot Analyses of Protein Extracts

Proteins of relevance were analyzed by Western blot as previously described [17]. Chemiluminescence signals were scanned with the imaging system Fusion FX6 Edge V0.7 (Vilber Lourmat, Eberhardzell, Germany). To quantify the signals, peak volumes of the protein-specific bands (≥five replicates) determined with EvolutionCapt Edge software (Version 18.12; Vilber Lourmat) were first standardized in relation to those of β-actin in the very same sample, and signals were normalized to those obtained from similarly processed control cells [12,17]. Because PLVAP is expressed only at very low levels by unchallenged iBREC, specific signals were normalized to those obtained from cells exposed to VEGF-A_165_. For the detection of faricimab, we used polyclonal goat antibodies directed against the γ-chain of an human IgG (Invitrogen, Thermo Fisher Scientific, #62-8420; 1:1000), and specific signals were set in relation to those obtained from cells exposed to VEGF-A_165_ and 1 mg/mL faricimab [17]. Data from multiple Western blot experiments performed with several independently prepared cell extracts were pooled and presented as graphs, or as scatter plots always containing means and standard deviations.

### 4.6. Statistical Analyses

The one-sample *t*-test, which takes the variation of the standard deviation into account although it appears to be zero, was used to compare antigen-specific Western blot signals from effector-treated cells to the hypothetical value of 1.00 of normalized signals. To compare several groups of antigen-specific Western blot or ELISA signals from differently treated cells, the one-way analysis of variance (ANOVA) followed by Dunnett’s multiple comparisons test was used. Data obtained by cell index measurements were analyzed with the two-way ANOVA, followed by Šidak’s multiple comparison test. Differences resulting in *p*-values below 0.05 were considered significant. All statistical analyses were performed with GraphPad Prism 9.4.1; means and standard deviations are provided as numbers, graphs or in scatter plots.

## 5. Conclusions

Dual binding of VEGF-A and Ang-2 by the bi-specific antibody faricimab lowered the secretion of Ang-2, and most importantly, reverted VEGF-A_165_-induced barrier impairment. Low cell index values and altered expression of proteins regulating para- and transcellular flow were normalized, although stabilization was only maintained with high concentrations of faricimab. The clinical significance of these findings for patients with macular edema lies in the ability of faricimab to not only counteract direct consequences of VEGF-A_165_ but also protect against the detrimental effects of Ang-2.

## Figures and Tables

**Figure 1 ijms-26-04318-f001:**
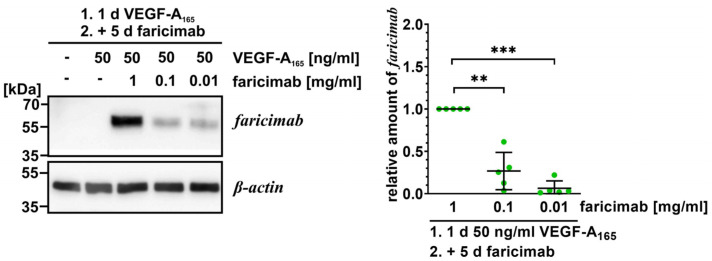
Faricimab is taken up by VEGF-A_165_-exposed iBREC. The VEGF antagonist was added to VEGF-A_165_-exposed iBREC before cells were harvested five days later for preparation of cell extracts and subsequent Western blotting. Specific signals were normalized to those obtained from cells treated with VEGF-A_165_ and 1 mg/mL faricimab. Pooled data of several Western blot experiments were analyzed as described in Section 4.6, and are shown as scatter plots with means and standard deviations. ** *p* < 0.01, *** *p* < 0.001; only statistically significant differences are marked. Original images are shown in Appendix A. Even after extended incubation, internalized VEGF antagonist can still be detected in a concentration dependent manner.

**Figure 2 ijms-26-04318-f002:**
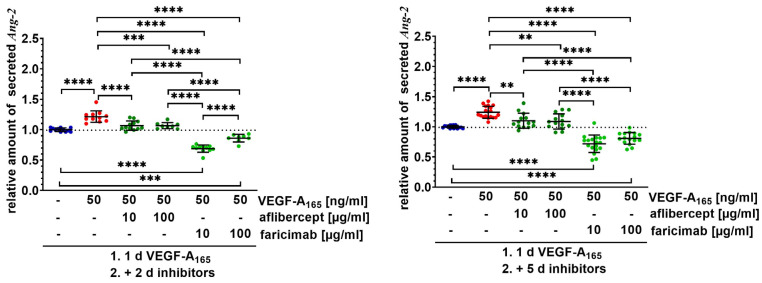
Faricimab sustainably counteracted higher secretion of Ang-2 by VEGF-A_165_-exposed iBREC. Confluent monolayers of iBREC were treated with VEGF-A_165_ for one day and subsequently with aflibercept or faricimab for additional two or five days, before cell culture supernatants were collected for determination of Ang-2 by ELISA. Pooled Ang-2-specific signals were normalized, and analyzed as described in Section 4. ** *p* < 0.01, *** *p* < 0.001, **** *p* < 0.0001, indicating statistically significant differences only. Secretion of Ang-2 was significantly increased after treatment with VEGF-A_165_ for three or six days. Aflibercept efficiently normalized Ang-2 levels, but faricimab lowered amounts of Ang-2 significantly more markedly.

**Figure 3 ijms-26-04318-f003:**
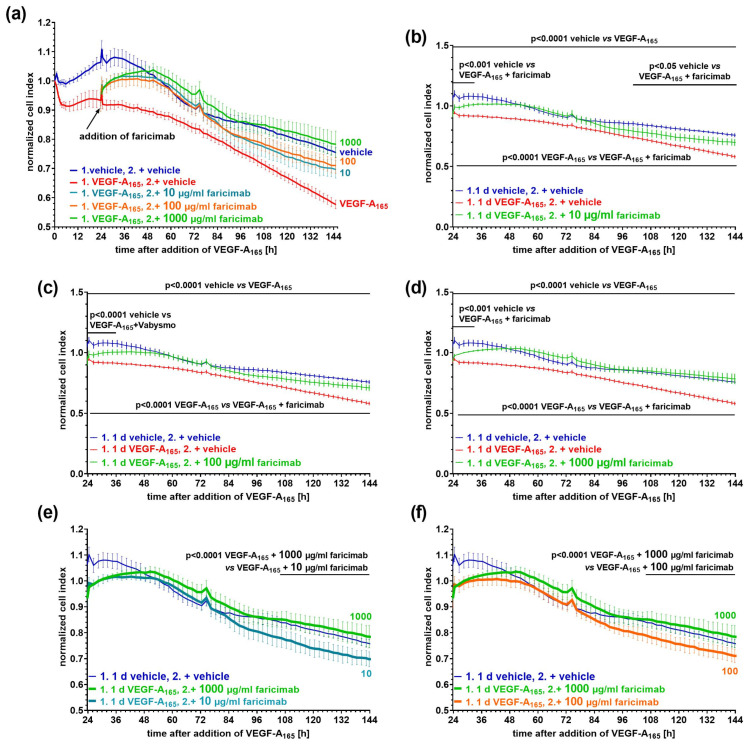
VEGF-A_165_-induced low cell index values were efficiently normalized after subsequent exposure to faricimab. Confluent monolayers of iBREC were treated with 50 ng/mL VEGF-A_165_ for one day before faricimab at final concentrations of (**a**,**b**,**e**) 10 µg/mL (n = 7), (**a**,**c**,**f**) 100 µg/mL (n = 8) or (**a**,**d**,**e**,**f**) 1 mg/mL (n = 6) was added, and the cell index was continuously determined. Control cells (n = 6) were exposed to vehicle only. Cell index values were normalized in relation to those measured just before the addition of VEGF-A_165_. Graphs show means ± standard deviations. (**a**–**d**) VEGF-A_165_ induced a significant and persistent decline of the cell index values. (**a**,**b**) After the addition of 10 µg/mL faricimab, low cell index values increased to normal levels within hours, but they only remained similar to those of control cells for about two days. (**a**,**c**,**d**) Normal cell index values were quickly re-established by (**c**) 100 or (**d**) 1000 µg/mL faricimab, and they did not significantly differ from those of control cells throughout extended exposure to faricimab. (**e**,**f**) 1000 µg/mL faricimab allowed for the most persistent stabilization compared to (**e**) 10 µg/mL and (**f**) 100 µg/mL faricimab.

**Figure 4 ijms-26-04318-f004:**
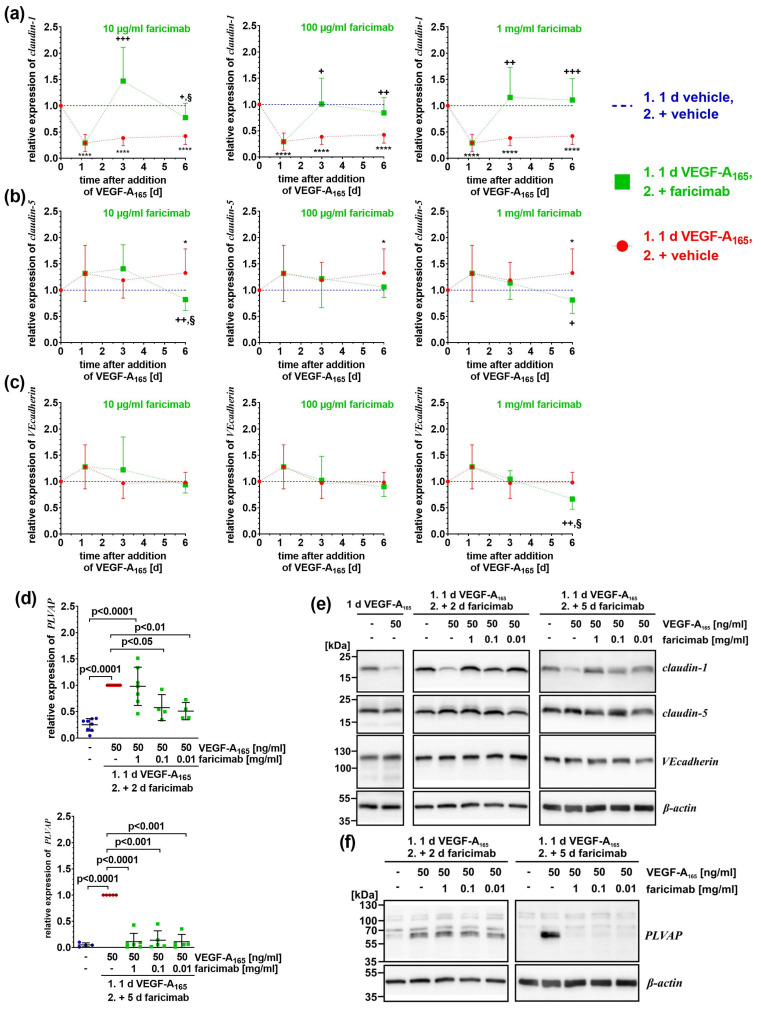
Faricimab reverted VEGF-A_165_-induced altered expression of claudin-1 and PLVAP. iBREC were pre-treated with 50 ng/mL VEGF-A_165_ for one day, before faricimab was added, and cells were harvested for preparation of cell extracts and subsequent Western blotting at indicated time points. (**a**–**c**) Specific signals of Western blot analyses (N ≥ 5 per condition and time point) were normalized as described in Section 4. Pooled data are shown as graphs with means and standard deviations. VEGF-A compared to control: * *p* < 0.05, **** *p* < 0.0001; VEGF-A compared to VEGF-A + faricimab: ^+^ *p* < 0.05, ^++^ *p* < 0.01, ^+++^ *p* < 0.001; VEGF-A + faricimab compared to control: ^§^ *p* < 0.05; only statistically significant differences are marked. (**a**) VEGF-A-induced low expression of claudin-1 was efficiently normalized by ≥ 100 µg/mL faricimab two and five days after the antagonist’s addition. Reversion of claudin-1 expression by 10 µg/mL faricimab was only achieved early after its addition. (**b**) Claudin-5 amounts increased over time, lowered again by faricimab. (**c**) Only extended exposure to VEGF-A_165_ followed by 1 mg/mL faricimab resulted in significantly less VEcadherin. (**d**) The band at ~60 kDa represents the PLVAP-specific signal, the faint bands at 70 kDa and ~100 kDa are unspecific [12]. Specific Western blot signals were normalized to those obtained from VEGF-A-exposed cells and pooled data—analyzed as described in Section 4.6—are shown as scatter plot with means ± standard deviations. Only prolonged treatment with faricimab was sufficient to completely down-regulate the VEGF-A_165_-induced expression of PLVAP. (**e**,**f**) Typical images of Western blot analyses of which original images are shown in Appendix A.

**Table 1 ijms-26-04318-t001:** Primary and secondary antibodies.

Target	Host, Type and Conjugate	Source ^a^	Working Concentration
actin	mouse, monoclonal	clone 5J11, Novus Biologicals, #NBP2-25142	700 ng/mL
β-actin	mouse, monoclonal	clone BA3R, Invitrogen, #MA5-15739	100 ng/mL
claudin-1	rabbit, polyclonal	Invitrogen, #51-9000	250 ng/mL
claudin-5	rabbit, polyclonal	Invitrogen, #34-1600	100 ng/mL
PLVAP ^b^	rabbit, polyclonal	Invitrogen, #PA5-110183	2 µg/mL
VEcadherin	rabbit, polyclonal	Cell Signaling Technology B.V., #2158	1:2000
whole rabbit IgG	goat, polyclonal, coupled to HRP	Biorad, #170-5046	1:15,000
whole mouse IgG	goat, polyclonal, coupled to HRP	Biorad, #170-5047	1:30,000

^a^ Biorad, Munich, Germany; Cell Signaling Technology B.V., Frankfurt, Germany; Invitrogen via Thermo Fisher Scientific, Schwerte, Germany; Novus Biologicals via bio-techne, Wiesbaden, Germany. ^b^ HRP, horseradish peroxidase; PLVAP, plasmalemma vesical-associated protein; VEcadherin, vascular endothelial cadherin.

## Data Availability

The original data used to support the findings of this study are either included in the article or available from the corresponding author upon request.

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
