# Peer review of "Faricimab Reverts VEGF-A_165_-Induced Impairment of the Barrier Formed by Retinal Endothelial Cells"

_ijms, 2025, doi:10.3390/ijms26094318_

Round 1

Reviewer 1 Report

Comments and Suggestions for Authors

This study by Jung et al investigates if dual binding of both growth factors VEGF-A and Ang-2 by the bi-specific IgG faricimab efficiently and sustainably reverses VEGF-A165-induced changes of the barrier formed by immortalized microvascular endothelia of the bovine retina (iBREC).  The paper is overall well written and the group has considerable expertise in research of retinal endothelial cells,  I have following major concern:

Lack of Faricimab only controls. The results of this manuscript showed that Faricimab is taken up by iBREC particularly at high concentrations of 1mg/ml.  Faricimab only controls are needed to exclude following possibilities: 1. Faricimab is taken up by both VEGF-A165-exposed and non-VEGF-A165  iBREC; 2.  internalized Faricimab could increase expression of claudin-1 and claudin-5 and subsequent increased CI value, independent of neutralization of VEGF-A165 and Ang-2.

Reviewer 2 Report

Comments and Suggestions for Authors

Comments for authors:

The manuscript entitled “Faricimab reverts VEGF-A165-induced impairment of the barrier formed by retinal endothelial cells”  by Dominik M.Jung and et al. is an interesting research, well written and well planned. The authors present an interesting compilation of information on the protective effect of Faricimab, which sustainably reverts VEGF-induced barrier impairment and protects against detrimental actions of Ang-2 by lowering its secretion.

  1. The significance of the findings has been explicitly mentioned in the title.
  2. The hypothesis was mentioned in introduction; therefore, it makes clear for reader in regard to the purpose of the study.
  3. The introduction is well written, including comprehensive data about the analyzed topics.
  4. Accuracy of the experimental design, methods and statistical analysis is well prepared and project. The methodology is well-written and includes many details that are important for its replicability and/or reproducibility.

This is a set of interesting and important data, providing scientific support for the argument that faricimab may have a protective role of the antagonist in early and late stages of diabetic retinopathy.

Reviewer 3 Report

Comments and Suggestions for Authors

This manuscript investigates the potential role of faricimab to reverse the VEGF-A165-induced breakdown of the retinal endothelial barrier using immortalized bovine retinal endothelial cells (iBREC) as a model. It presents in vitro experiments demonstrating how faricimab restores barrier function, modulates protein expression (claudin-1, claudin-5, PLVAP), and reduces Ang-2 secretion. It addresses an important topic and generates findings of clinical interest. With the minor revisions (as below), particularly those clarifying translatability and conflicts of interest, this manuscript would be suitable for publication.

  1. Figure 3 currently includes redundant panels displaying both means and mean ± SD plots, which may be confusing for readers. Consider consolidating these views or highlighting only the key comparison (e.g., CI normalization across different faricimab concentrations). Clearer labeling and grouping of time points would also improve interpretability.
  2. Figures 4 and 5 both present protein expression data relevant to barrier integrity. Consider merging these into a single composite figure categorized by tight junction vs. transcellular transport components to enhance clarity and reduce redundancy
  3. It would strengthen the study to include immunofluorescence staining of claudin-1 and claudin-5 to confirm their junctional localization and visually support the Western blot findings. This would enhance the physiological relevance of your observations on barrier restoration.
  4. The use of bovine cells, while justified, still limits direct extrapolation to human in vivo responses. More discussion on interspecies differences and relevance to clinical outcomes is warranted. A short section comparing iBREC data to primary human REC or co-culture models would enhance robustness. BTW, both hREC and hChEC cells can be purchased from ATCC.

Round 2

Reviewer 1 Report

Comments and Suggestions for Authors

The concerns have been addressed. No further questions